# Community Research Fellows Training Program: Evaluation of a COVID-19-Precipitated Virtual Adaptation

**DOI:** 10.3390/ijerph20043254

**Published:** 2023-02-13

**Authors:** Nicole Ackermann, Sarah Humble, Jacquelyn V. Coats, Carlette Lewis Rhone, Craig Schmid, Vetta Sanders Thompson, Kia L. Davis

**Affiliations:** 1Division of Public Health Sciences, Department of Surgery, School of Medicine, Washington University, St. Louis, MO 63110, USA; 2Brown School, Washington University, St. Louis, MO 63110, USA; 3Community Research Fellows Training, School of Medicine, Siteman Cancer Center & Washing University, St. Louis, MO 63110, USA

**Keywords:** community engagement, stakeholder engagement, health equity, engagement science

## Abstract

Community engagement is important for promoting health equity. However, effective community engagement requires trust, collaboration, and the opportunity for all stakeholders to share in decision-making. Community-based training in public health research can build trust and increase community comfort with shared decision-making in academic and community partnerships. The Community Research Fellows Training (CRFT) Program is a community-based training program that promotes the role of underserved populations in research by enhancing participant knowledge and understanding of public health research and other relevant topics in health. This paper describes the process of modifying the original 15-week in-person training program to a 12-week online, virtual format to assure program continuation. In addition, we provide program evaluation data of the virtual training. Average post-test scores were higher than pre-test scores for every session, establishing the feasibility of virtual course delivery. While the knowledge gains observed were not as strong as those observed for the in-person training program, findings suggest the appropriateness of continuing to adapt CRFT for virtual formats.

## 1. Introduction

Our previous research suggests that as science advances and health research and care become more complex, it remains important that patients and other health stakeholders have the ability to participate in decision-making at the level desired [1,2]. To facilitate outcomes acceptable to stakeholders, innovative strategies are needed to ensure that patients, family members, and other stakeholders have sufficient knowledge of research and research methods to confidently share their preferences, needs, concerns, and priorities [1,2]. Lack of knowledge or uncertainty about relevant issues may result in a failure to engage fully in research and/or patient decision-making processes [1]. This lack of engagement increases the likelihood of deference to clinical professionals, the health system, friends, family, and social media influences, ultimately heightening the likelihood of succumbing to misinformation and/or dissatisfaction with health decisions [1]. Thus, researchers and clinicians committed to community engaged research and practice must provide opportunities for learning that increase research knowledge and literacy [2,3].

Community engagement is an important tool for promoting positive social and community health change [4]. However, effective community engagement requires a process that builds trust, values contributions of all stakeholders, and generates a collaborative framework [5]. Community-based training in public health research can contribute to this process and allows academics to build capacity of community members to engage in community-based participatory research (CBPR), a collaborative form of community engagement [2,3,6,7]. Researchers are increasingly recognizing the role that CBPR can play in research designed to reduce health disparities and move toward health equity [8,9,10,11,12,13,14,15,16]. When communities and researchers are interested in and prepared for engagement, community engaged research can result in the identification and examination of health related concerns and conditions of relevance to the community [17,18,19]. Increasing community engaged research generally, and CBPR specifically, are viewed as research strategies capable of improving the acceptability and effectiveness of health promotion and disease prevention activities as well as research itself [15,20,21].

The Community Research Fellows Training (CRFT) Program is a community-based training program that promotes the role of underserved populations in research by enhancing the capacity for CBPR [2,3,6,7]. This 15-week training program was adapted from the Community Alliance for Research Empowering Social Change (CARES) program. CRFT includes topics relevant to a wide range of health research [22]. The advisory group that designed and implemented CRFT reviewed the CARES materials and determined what program components were appropriate for the urban, Midwest location of the CRFT program. Recent CRFT cohorts have addressed 18 topics presented in 15 weekly sessions, each lasting 3 h. Topics include health disparities, health literacy, ethics, cultural competency, epidemiology, quantitative and qualitative research methods, chronic disease prevention, clinical trials, study design, program evaluation, and grant writing. Sessions were designed to be lay-friendly, while also consistent with Master of Public Health (MPH) curriculum. The Principal Investigator recruited one or two faculty members to lead each program session, depending on session content. The faculty selected are experts in the field, teach and perform research in these areas, and have strong community connections. Faculty are from multiple institutions in the region. The training uses multiple teaching approaches (large didactic interactive lectures, small group activities, group exercises, and small and large group discussions) to explain topics in ways that accommodate a variety of learning styles [23]. Other pedagogical elements and principles are available in previous evaluation studies [2,3].

The inclusion criteria for the CRFT program required participants to be at least 18 years old and live or work in, or willing to commute to, the St. Louis greater metropolitan area. The program participants do not receive compensation but receive free training and resources. Participants are referred to as “Fellows” to further empower and engage them in the academic process.

In April 2020, a sixth CRFT cohort was cancelled due to the COVID-19 shutdown. The cancellation occurred after the application deadline but prior to any formal acceptance of Fellows into the program. This paper describes the process of modifying the original 15-week in-person program to a 12-week online program offered in a virtual format, to assure program continuation. The team decided that the virtual program was the most feasible strategy to continue teaching community members during the COVID-19 pandemic.

The Human Research Protection Office at Washington University in St. Louis classified the study of “virtual” CRFT underlying this paper as program evaluation and non-human subject research.

## 2. Methods

### 2.1. Community Research Fellows Training Program

A community advisory board (CAB) that consists of 12 members, 8 CRFT graduates and 4 community stakeholders, helps to guide all aspects of the CRFT program, including recruitment, selection, program implementation, and evaluation of the program. The CAB and Project Team collaborated as a revision team to consider how to continue an established and evaluated community-based training program [2,3,6,7] in a virtual format. The focus of revision team discussions involved how to avoid the fatigue and attention span deficits inherent in a virtual format as well as highlighting the best practices for using virtual technology platforms like Zoom© (Zoom Video Communications, Inc., San Jose, CA, USA) and Canvas© (Instructure, Inc., Salt Lake City, UT, USA) while maintaining fidelity with the quintessential elements of in-person CRFT. These modifications included reducing the length and number of sessions, creating engaging activities in a virtual format, innovation in community engagement homework, creating networking opportunities, and changing program evaluation procedures.

Discussions on transitioning CRFT to a virtual delivery began in September 2020 after several members of the Project Team had adapted other courses to a virtual format for the academic summer and fall semesters. They were able to bring lessons learned from those experiences to the discussions. In addition, the Washington University in St. Louis Center for Teaching and Learning provided many resources [24,25,26] to faculty and staff at the university.

### 2.2. Goals

The program continues to pursue the original goals of the CRFT program. The first goal is to equip community members with sufficient research knowledge to be good consumers of that research. In addition, the program seeks to help community members to understand how to use research to improve the community health status and well-being. The final goal is to increase community readiness for community/academic collaboration and partnership on mutually beneficial projects and programs. Additionally, the same multidisciplinary faculty taught the virtual program who had taught the in-person program. The team also recommended a reduced number of sessions (from 15 to 12), inclusion of an online orientation that provided an overview of Canvas© and Zoom© navigation, and one workshop intended to train community members in qualitative research methods, as well as an online graduation ceremony and optional outdoor, in-person celebration (See Table 1).

### 2.3. The Online Format

The Project Team received permission from the home institution to administer the course using the Canvas© Learning Management System (LMS). The session readings, activities, homework, and supplemental materials were available in Canvas© modules. The Project Team decided that each lecture topic would be 45 to 50 min, with 10 to 15 min reserved for questions. Faculty modified the session objectives and class activities to accommodate the new 2 h course duration (See Table 1). The Project Team and volunteers facilitated and monitored activities completed in Zoom© breakout rooms because small group activities were included in the original in-person CRFT format. There was agreement that homework should be continued but modified to accommodate ongoing physical distancing which was practiced as a pandemic-spread mitigation measure in the region.

Given the online format, the Project Team made the decision to keep the virtual online cohort (Cohort VI for 2021) small to encourage greater interactive participation. The cohort was limited to a maximum of 25 participants, which was similar to Cohorts III–V. Prior CRFT cohorts were recruited through a local newspaper, E-mail, community websites, community newsletters, flyers, personal referrals, and word of mouth. However, the Project Team decided to approach those who applied for the cancelled 2020 cohort (n = 29), individuals who contacted staff to express interest in a future cohort (n = 23), and send emails to the CRFT alumni network and other CRFT affiliated individuals. Applicants had an opportunity to use a previous application, provide an updated application, or submit a new application.

The Project Team reviewed the applications and selected a cohort that was diverse in work, education, and life experiences, just as the in-person CRFT cohorts had been. Twenty-three of the thirty-eight individuals who applied to participate in the 2021 cohort had previously applied in 2020. Sixteen of those twenty-three past applicants were selected as Fellows for the 2021 cohort. There were 15 new applicants in 2021, with 9 selected to participate in the 2021 cohort.

### 2.4. Canvas© Training

We offered training on how to use Canvas©, the learning management system during the CRFT program orientation. Canvas© was mainly used as a way to store and organize CRFT program materials, whereas assessments were given outside of Canvas©, using a survey administration platform, REDCap^®^. REDCap^®^ is a secure, web-based software platform designed to support data capture for research and evaluation. The orientation session lasted 2 h, with about 30 min set aside for a walkthrough of virtual course materials, including how to use Canvas©. All 25 Fellows were able to attend the orientation session.

During the introduction to Canvas©, CRFT team members demonstrated going to the institution specific Canvas© website, creating an account, logging into the system, and navigating through the course materials, which were set up using the modules feature in Canvas©. In addition to the live walkthrough of Canvas©, the general and institution specific information was sent via email. The process of creating an account was more difficult for Fellows than using Canvas©, as several Fellows reached out to the CRFT team for assistance and questions on account creation. However, once an account was created, Fellows did not voice concerns on accessing course materials. One Fellow, who also did not finish the program, was never able to create a Canvas© account. Fellows also received a one-page guide on how to access Canvas© prepared by the CRFT team.

### 2.5. Assessment of Participant Knowledge

Fellows’ increase in knowledge of public health principles and CBPR was assessed via a questionnaire administered before session 2 (baseline) and after session 12 (final) of the program. Each questionnaire to assess public health knowledge consisted of 20 identical multiple-choice questions created by the CRFT Project Team and faculty to measure the Fellow’s knowledge of the CRFT training topics and learning objectives (Table 1). The questions selected for administration were related to the learning objectives the faculty member(s) intended to cover during the weekly session. Fellows’ baseline and final scores were linked using their personally selected CRFT ID numbers.

A baseline and final score were computed for each individual by summing the responses for all questions, where a value of 1 was assigned for a correct response and a value of 0 for an incorrect response (including missing and “I don’t know” responses). A maximum total score of 20 points was possible for each score. We then converted this to a percentage score. To assess the difference between the baseline and final assessment, we performed a non-parametric test, the Wilcoxon signed-rank test, due to violations of normality assumptions.

### 2.6. Pre- and Post-Tests

Fellows took pre- and post-tests at each of the 12 sessions. Pre- and post-test measures assessed the content at the beginning of each session and at the end of each session, using identical items. All tests were shortened from the pre- and post-tests administered during the in-person sessions. The tests were shortened from 10 items to 5 items in order to reduce participant burden. Pre- and post-tests were scored similar to the baseline and final assessment, using a sum score of correct responses, where a value of 1 was assigned for a correct response and 0 for incorrect response (including missing or don’t know). A maximum total score of 5 was possible for each pre- and post-test. We then converted this to a percentage score. Due to the violation of normality assumptions, we used a non-parametric test, the Wilcoxon signed-rank test, to evaluate the score differences on pre-test and post-test for each session.

### 2.7. Evaluation of Fellow Satisfaction

At the end of each of the 12 training sessions, Fellows completed session evaluations. Fellows rated their satisfaction level on that week’s topic and content covered as well as evaluation of the faculty member presenting. Evaluations consisted of six statements with Likert-scale response options. The Likert-scale items were about satisfaction with the learning objectives for the topic as well as the perceived quality of the presentation for the session. In addition, there were four free response questions.

Analyses were completed using SAS/STAT 9.4 (SAS Institute, Cary, NC, USA).

## 3. Results

There were 25 Fellows enrolled in the virtual online CRFT program. Of these 25 Fellows, 23 (92%) completed the 12-week virtual training program. We obtained complete baseline and final assessment data from 21 of the 23 Fellows who completed the program (91%). Table 2 provides demographic information on the full cohort (n = 25).

The three largest groups comprising the cohort were individuals affiliated with community-based organizations (32%), followed by health care workers (28%) and individuals with multiple roles (28%). The remaining Fellows were community members and those affiliated with faith-based organizations (8%) and those in academia (4%). The mean age of Fellows was 45, with an age range from 25 to 65. All Fellows had some college or an associate’s degree or higher, including 48% with graduate education, and 68%of the cohort reported that they had previously completed a course on research. The majority of fellows were African-American/Black (76%).

All of the Fellows who completed both the program and final assessments (n = 21) were female and 71% were African-American/Black. The four individuals who did not complete the training and/or the final assessment were African-American/Black, with one male and three females. These individuals were similar to the 21 Fellows who completed the program and final assessment in all respects except age. The mean age was slightly lower, 38 and ranged from 33 to 43, compared to the 21 individuals who completed the program and final assessments (mean age = 46.8, ranging from 25 to 65).

### 3.1. Assessment of Participant Knowledge

The average score increased from 65% correct on the baseline public health and CBPR knowledge assessment to 76% correct on the final assessment (n = 20 (one outlying observation excluded), mean change of 11%, range −10% to 40%). Only two Fellows (out of 21, 10%) decreased their scores from baseline to final assessment. The Wilcoxon signed-rank test for this knowledge change was significant (*p* = 0.0005).

Due to the online nature of the course, participants found it challenging to complete pre-/post-tests. In addition, due to non-participant technical errors involving REDCap^®^, approximately halfway through the course, Fellows did not always receive reminder emails to take the test. Overall, four Fellows (16%) completed all pre-/post-tests. During the first half of the course, 60–68% of participants completed the pre- and post-tests, and during the second half of the course, 36–54% of participants completed both tests.

There is evidence of knowledge gained. However, the in-person program had a greater number of sessions with Fellows showing statistically significant gains compared to the online program [2,3]. Data in Table 3 indicates that although average post-test scores were higher than pre-test scores for every session, only Session 1—Public Health and Health Disparities (17.7), Session 3—Health Literacy (11.3), Session 4—Introduction to Epidemiology (25), and Session 6—Evidence-based Public Health and Program Planning (24) had a statistically significant difference in pre- and post-test scores. An examination of the pre-test scores indicates that three of these sessions—Introduction to Epidemiology (29.6), Evidence-based Public Health and Program Planning (44), and Health Literacy (47.2)—were among the areas of greatest knowledge deficit. Sessions 2—Community Based Participatory Research & Community Organizing, Session 5—Behavioral Health/Cultural Competency, and Sessions 7 through 12—Research Methods & Data, Quantitative Methods, Qualitative Methods, Clinical Trials & Biobanks/Research Ethics, Health Policy Research/Grant Writing, and Human Subjects Certification—did not have a statistically significant difference in pre- and post-test scores.

### 3.2. Participant Feedback

The following quotes were offered in the final assessment in response to the prompt to “Please provide us with any additional comments or suggestions”. The Fellows indicated that they appreciated the effort to continue the program despite ongoing pandemic precautions. Representative quotes are provided below.

“Thank you for taking the time to re-structure the program and your continued energy, patience and support throughout the program!”

“Thank you for the opportunity to participate in this outstanding program. I (sic) is a testament to the dedication of your staff to engage the community in the advancement of public health. Please continue the program!”

“This was an awesome opportunity. The information gained will be used to help me to be a better collaborator as I work to improve areas of the community in which I serve”.

In addition, Fellows were asked to share opportunities to change or improve the training sessions. There were comments related to virtual versus in person sessions. Some Fellows liked the virtual sessions and some did not.

“I think in person would of course be better, but with the pandemic that couldn’t be done”.

“I really liked the virtual setting”.

“Do not offer online/virtually”.

However, the main concerns discussed were the desire for more interaction and group discussions, more time for each session (“Continue to offer virtual classes, extend the time 30 min”).

Finally, data from the final assessment indicated that 81% (n = 17) of Fellows agreed or strongly agreed “the structure of the training was beneficial to the learning process”. In addition, 95% (n = 20) agreed or strongly agreed that they “would recommend the CRFT program to others”.

## 4. Discussion

This paper describes the process of modifying the original CRFT 15-week in person training program to a 12-week virtual online format. The virtual format proposed was the most feasible strategy to continue training community members during the COVID-19 pandemic. In addition, it increased the number of communities and participants with access to this community-based public research training program [2,3,6,7] by continuing the training during the COVID-19 pandemic.

There were both facilitators and barriers to implementation. Throughout the 12-week training, Fellows remained engaged in the training, indicating high potential for a virtual CRFT training environment. In addition, all CRFT faculty continued to be committed to the goals of the training and willing to adapt curricula from for online virtual format. Faculty used several Zoom© features, including breakout rooms, polls, and the chat to facilitate participation. On the other hand, while many Fellows understood and supported a virtual format, feedback also indicates that many value opportunities for face-to-face trainings. In addition, the recommendations offered suggest the need to add additional time to online sessions. Low pre-/post-test completion rates indicate some limitations in administration of online materials asynchronously, as discussed in the next section. One participant who failed to complete the program was never able to establish an LMS account and this likely hampered the quality of the experience and program completion.

The cohort of Fellows was comprised of individuals affiliated with community-based organizations and other diverse stakeholders. The cohort had greater age and educational diversity than racial/ethnic and gender diversity. Ninety-two percent of CRFT VI Fellows completed the 12-week training program and ninety-one percent of these Fellows completed both the baseline and final assessments. There was a statistically significant average score increase from the baseline assessment to the final assessment (mean change of 11%).

The average post-test scores were higher than the pre-test scores for every session; however, as Table 3 data indicate, only four sessions—Public Health and Health Disparities, Health Literacy, Introduction the Epidemiology, and Evidence-based Public Health and Program Planning—had a statistically significant difference in pre- and post-test scores. An examination of the pre-test scores indicates that three of the four sessions (Health Literacy, Introduction the Epidemiology, and Evidence-based Public Health and Program Planning) are important areas for public health and research and were among the areas of greatest knowledge deficit.

Given the virtual nature of the course, participant evaluations are important. Ninety-five percent of Fellows indicated that they would recommend the CRFT program to others and a smaller, but large, percentage reported that the structure of the program was beneficial to the learning process. While these findings establish the feasibility of virtual course delivery, the knowledge gains were not as strong as those observed for the in-person training program.

### Limitations

Despite these positive findings, there are limitations to this evaluation and some findings should be interpreted with caution. First, a low percentage of Fellows completed all of the pre/post-tests (16%), and the percentages completing the tests during the first half of the program was significantly higher than the percentage completing the tests in the second half of the program. One issue was the ability to send reminders to Fellows reliably. With implementation of the virtual format, CRFT staff reminded Fellows to complete pre- and post-tests at the beginning and end of each session. The in-person format allowed staff to ensure compliance with the tests, while the effort to use REDCap^®^ to provide reminders failed. We learned that it is important to test all features of the LMS system and develop alternative strategies for implementation of program components prior to the launch of virtual programming, including those related to evaluation processes.

While prior CRFT cohorts have been predominantly female [2,3], this cohort included fewer males than the previous cohorts. There are 29 male CRFT alumni (19% of alumni). Cohort V was 25% male (unpublished data). Lower participation by males in this cohort may have been due to differences in COVID work roles and schedules as well as family and domestic adjustments among male and female workers. It may also be important to understand how the restricted recruitment strategy created by COVID, differential recruitment efforts of alumni, and the use of the virtual online format affected male participation. Similarly, while there was educational diversity, there were no Fellows with a high school education or less who completed the program. To date, there are only four CRFT alumni with a high school education or less. These data speak to the need to consider programming that may appeal to individuals with less education. This may require programming that provides similar content but offered through a series of focused short courses that provide more time and support for participants to absorb the information than is available in the current CRFT program. Despite these limitations, the findings suggest the appropriateness of continuing to adapt CRFT for virtual formats.

## 5. Conclusions

CRFT VI demonstrated the feasibility of providing community-based training in public health research using a virtual online format. Despite the existence of other programs designed to provide community training on research [27,28], most formats reported on are in person. This virtual adaptation was conducted with fewer sessions, sessions were shorter in duration, and homework assignments and activities were modified to accommodate the virtual format. Although there were a greater number of sessions with statistically significant gains for the in-person program compared to the virtual online program, Fellows demonstrated gains in research knowledge in both formats. The virtual implementation experience highlights the need for adequate staff support for participants, as well as for creating and managing online course content. Findings also highlight the importance of participant training on the use of the course management system and the virtual meeting platform prior to program implementation to increase comfort and familiarity. These findings suggest the potential of a virtual online community research training program, although adjustments and further evaluation are necessary.

## Figures and Tables

**Table 1 ijerph-20-03254-t001:** Cohort VI Virtual Online Session Topics and Learning Objectives.

Topics		Learning Objectives
		* Included in the in-person curriculum; ^+^ Combined session
Session 1	Public Health Research & Health Disparities *	
		Define public health research
		Identify and explain types of research
		Explain why research is important
		Define health disparities
		Identify major health disparities in the St. Louis including those by gender, race/ethnicity, geographic location, and socioeconomic status
		Understand and provide example of causes of health disparities with respect to prevention, incidence, and mortality
		Discuss the social determinants of health
		Describe public health strategies and interventions for reducing health disparities
**Activity:** Discussion of video “Segregation by Design” All; 30 min		
Session 2	Community Based Participatory Research & Community Organizing *^+^	
		Describe history and principles of CBPR
		Critically evaluate their own position within their community(ies) and their potential roles within CBPR projects
		Describe methods to ensure that CBPR research benefits all partners
		Lessons learned from CBPR projects
		CBPR efforts in St. Louis
		Describe history and principles of community organizing
		Describe community organizing resources useful for public health initiatives
		Identify and develop relevant well-framed community organizing strategies
Activity: Planning the first full meeting of a CBPR partnership 4 Breakout Rooms (4–5 participants); 15 min		
Session 3	Health Literacy *	
		Define health literacy
		Understand the limited literacy perspective
		Describe the association between literacy and health
		Describe health literacy on a national scale
		Discuss current research on health literacy
Activity: Health literacy and nutrition label analysis 4 Breakout Rooms (4–5 participants); 25 min Homework 1: Social determinants of health		
Session 4 Introduction to Epidemiology *		
		Define epidemiology
		Identify major contributions of epidemiology
		Identify frameworks for understanding disease processes
		Compare and contrast observational studies vs. clinical trials
Activity: Zombie outbreak activity—disease detectives All; 30 min		
Session 5 Behavioral Health/Cultural Competency *		
		Define behavioral health
		Identify the role that behavioral health plays in physical health
		Describe efforts to integrate behavioral and physical health
		Define cultural competency
		Describe the need for culturally competent research and practice based on a historical perspective
		Identify contributing risk factors for health disparities
		Identify skills associated with culturally competent practices
Activity: “Why is it important to talk about race?” Poll tool to keep track of responses 4 Breakout Rooms (4–5 participants); 25 min Homework 2: Windshield Survey		
Session 6 Evidence Based Public Health & Program Planning *		
		Define evidence based public health
		Develop SMART goals for programs and projects
		Identify culturally competent evaluation approaches
		Understand the importance of evaluation
Activity: Create a logic model for a new culturally competent diabetes program 4 Breakout Rooms (4–5 participants); 25 min		
Session 7	Research Methods & Data *	
		Define research
		Describe the steps of the research process
		Identify and explain research methodology
		Identify appropriate research methods and techniques
		Define data
		Compare and contrast quantitative and qualitative data
		Compare and contrast primary data and secondary data
Activity: Identify the appropriate research method and data source for scenarios 4 Breakout Rooms (4–5 participants); 25 min Homework 3: Grocery Store Audit		
Session 8	Quantitative Methods *	
		Identify strengths and weakness of quantitative methods
		Describe strengths of mixed methods approaches
		Describe stages of questionnaire design
		Identify sampling methods
		Understand usefulness of statistics in health research
		Understand p-values and odds ratios
Activity: Analyze different soda brands to determine position on diet sodas 4 Breakout Rooms (4–5 participants); 30 min Homework 4: Photovoice		
Session 9	Qualitative Methods *	
		Define basic principles of qualitative research methods
		Describe the characteristics of qualitative research
		Describe the advantages and disadvantages of qualitative methods
		Understand and distinguish between different types of qualitative approaches
		Understand focus groups and Photovoice qualitative research methods
		Understand the relationship between qualitative and quantitative research methods
		Discern when a qualitative research design is desirable
Activity: Photovoice—full group discussion of select photos taken by cohort addressing how social capital impacts the helath of their community All; 30 min		
Session 10 Clinical Trials & Biobanks/Research Ethics *^+^		
		Understand clinical trials research
		Describe the role of clinical trials research in advancing medical practice
		Discuss the impact of minority participation in clinical trials research
		Define bio-repository and describe the type of research conducted from bio-repository data
		Discuss the risks and benefits of minority participation in bio-repositories
		Define research ethics and bioethics
		Compare and contrast clinical ethics vs. research ethics
		Identify examples of unethical practices in research
		Understand ethical theories and professional ethical duties
		Identify historical milestones in ethics
		Understand the Belmont Report
		Understand NIH–IRB Protocol Review Standards
Homework 5: Quantitative Methods		
Session 11 Health Policy Research/Grant Writing *		
		Define health policy and health services research
		Identify and develop relevant well framed health policy research questions
		Describe public use and other common data sources for health policy research
		Understand grant guidelines and requirements
		Understand the power of collaboration for grant writing
		Develop SMART goals and specific aims
		Understand components of a good grant proposal
Homework 6: Final Photovoice		
Session 12 Human Subjects Certification *		
		Participants will be certified in the conduct of human subjects research
		Conduct an informed consent process to recruit a participant in a research study
		Develop a humans subjects and HIPPA compliant research proposal
**Session Omitted** Family Health History		

**Table 2 ijerph-20-03254-t002:** Demographic characteristics of CRFT participants in evaluation sample (n = 25).

	n	%
Characteristics		
Sex		
Female	24	96
Male	1	4
Race/Ethnicity		
African American/Black	19	76
White	3	12
Other	3	12
Education Attainment		
Graduate Degree	12	48
Bachelor’s Degree	7	28
Some College/Associate’s Degree	6	24
Affiliation		
Academic	1	4
Community-based Organization	8	32
Community Member/Faith-based Organization	2	8
Healthcare Worker	7	28
Multiple Roles	7	28
Previously Taken a Research Course		
Yes	17	68
No	8	32
Age (years)		
Mean	45.0	
SD	12.9	

**Table 3 ijerph-20-03254-t003:** CRFT VI Virtual Online Pre-test and Post-test Scores.

CRFT VI Pre-Test and Post-Test Scores (Percent of Total Correct at Each Session)										
		Pre-Test Score			Post-Test Score			Score Difference (Post/Pre)		Wilcoxon Signed-Ranks Test
Sessions	n	Mean	SD	n	Mean	SD	n	Mean	SD	*p*
1. Public Health Research & Health Disparities	24	76.7	12.7	17	92.9	12.1	17	17.7	15.6	0.0015
2. CBPR & Community Organizing	23	70.4	18.9	16	85.0	15.5	15	9.3	21.2	0.1796
3. Health Literacy	25	47.2	21.5	16	60.0	19.3	16	11.3	17.8	0.0449
4. Intro to Epidemiology	25	29.6	21.7	16	56.3	27.5	16	25.0	25.8	0.0028
5. Behavioral Health & Cultural Competency	24	82.5	12.3	15	85.3	9.2	15	5.3	11.9	0.2188
6. EBPH & Program Planning	25	44.0	20.0	15	73.3	19.5	15	24.0	29.5	0.0068
7. Research Methods & Data	15	64.0	22.9	13	75.4	26.0	9	17.8	25.4	0.0859
8. Quantitative Methods	17	32.9	14.0	9	44.4	27.9	9	11.1	30.2	0.2148
9. Qualitative Methods	22	65.5	27.7	12	85.0	12.4	12	16.7	28.1	0.0849
10. Clinical Trials and Biobanks, Research Ethics	19	54.7	26.5	13	66.2	20.6	13	3.1	25.6	0.8359
11. Health Policy & Grant Writing	20	64.0	15.4	10	76.0	12.7	10	12.0	19.3	0.1563
12. Human Subjects Certification	17	49.4	20.2	11	58.2	22.7	11	12.7	24.1	0.1719
	Baseline Score			Final Score			Score Difference (Final/Baseline)			Wilcoxon Signed-Ranks Test
	n	Mean	SD	n	Mean	SD	n	Mean	SD	*p*
**Overall Course**	20	65.3	12.4	20	76.3	12.8	20	11.0	12.2	0.0005

## Data Availability

Not applicable.

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
