# Peer review of "Community Research Fellows Training Program: Evaluation of a COVID-19-Precipitated Virtual Adaptation"

_ijerph, 2023, doi:10.3390/ijerph20043254_

Round 1

Reviewer 1 Report

The text presents formatting errors revealing little care in its presentation.

The objective of the study, the description of the adaptation of the CRFTP to a virtual environment, is not accomplished.

Most of the information pertains to CRFTP in face-to-face training.

The introduction does not present studies on teaching in a virtual environment.

Description of the process of modifying the original CRFTP to an online program is limited to a reduction of sessions, contents, and items to be evaluated, besides noting some lack of knowledge in the use of virtual platforms such as Canvas.

The references are mostly outdated and are characterized by some inbreeding.

Sorry, but I am not able to understand the contribution of the present study to the research field.

Author Response

Thank you for your review of this paper and your comments and recommendations. We have responded to your comments and attempted to address your concerns. 

First, we acknowledge and apologize for the formatting errors that appeared in the version of manuscript that was sent out for review. These were not errors that appeared in the docment submitted. However, they were overlooked when reviewing the submitted document. We have worked to correct these errors, as well as errors in the references that were detected. 

We believe that this manuscript makes a contribution to the literature and have attempted to strengthen the content that describes the virtual adaptation. Specifically, we note that we used training and resources of the Center for Teaching and Learning at our institution as we planned the restructuring and revision of the course. We have provided these citations. We have added paragraphs discussing the training to facilitate creation of LMS accounts and use of the system complete homework. In addition, we have added information on when we used large group discussion, small group discussion in breakout rooms, and session activities to Table 1. 

This paper does reference the existing scholarship on the program developed to strengthen community ability to participate in community/academic partnerships. This work is ongoing and we attempt to share our learning with other scholars interested in community engagement. This has included a description of working with CRFT alum to conduct research important to them and their community, collaboration on a pilot of a youth version, and now the evaluation of a virtual cohort. We believe that scholars working in rural areas, or with working adults with families may find the virtual option more feasible to implement. Thus, we have sought to share information on the adaptation and its evaluation.  

Reviewer 2 Report

Authors have written an interesting article about important topic.

My main concern is that only "four Fellows (16%) completed all pre-/post-tests. During the first half of the course, 60%-68% of participants completed the pre- and post-tests, and during the second half of the course 36-54% of participants completed both tests."

Is this enough? Can this lead to some misinterpretations? Can resuls be generalised? How could we get rates higher? Please discuss.

Minor concerns:

Lines 32-44: Please make it clear that you are citing your own previous articles [1] and [2].

Please clarify: Were test questions before and after the program exactly same? And also before and after each module?

Please discuss: Why almost only women participated? How could you get also men to participate?

Line 178: 21 out of 25 is 84%. If authors want to use 91% it should be written out that it means 21 out of 23.

Author name "Carlette Lewis Rhone" is written with bigger font.

Extra spaces in text:

Line 20: extra space after "partnership"

Line 34: extra space after "desired"

Line 42: extra space after "decisions."

Line 51: extra space after "engagement [2-3. 6-7]."

Line 63: extra space after "program."

Line 101: extra space after "CRFT."

Line 110: extra space and dots after "program."

Line 112: extra space after "navigation."

Line 144: extra space after "pre- and"

Line 150: extra space after " the"

Table 1: extra space in "Session 10" after "Compare and contrast"

Line 198: extra space after "-10% to 40%)."

Line 203: extra space after "offering"

Line 256: extra space after "envorinment."

Line 257: extra space after "format."

Line 259: extra space after "trainings."

Line 288: extra space after "Funding:"

Line 289: extra space after "Medicine."

Line 291: extra space after "Statement:"

Line 295: extra spaces after "written" and "sheet."

References have multiple small formatting errors. Please correct.

Author Response

Thank you for your review of this paper and your comments and recommendations. We have responded to each comment and recommendation as indicated below. We believe that the edits made in response to your review have improved the manuscript. 

1. My main concern is that only "four Fellows (16%) completed all pre-/post-tests. During the first half of the course, 60%-68% of participants completed the pre- and post-tests, and during the second half of the course 36-54% of participants completed both tests."

Is this enough? Can this lead to some misinterpretations? Can resuls be generalised? How could we get rates higher? Please discuss.

We have addressed this concern in the limitations section of the paper. We acknowledge this as a limitation of the current program evaluation, and discuss how this was managed in previous cohorts. We also discuss the need to test the technology used to implement a virtual program, including features of the program used to collect evaluation data and features of the learning management system to improve pre-post-tests completion throughout the program.

Minor concerns:

2. Lines 32-44: Please make it clear that you are citing your own previous articles [1] and [2].

We have acknowledged and made it clear that lines 32-44 cite our previous work.

3. Please clarify: Were test questions before and after the program exactly same? And also before and after each module?

We have clarified that the questions asked before and after each module were exactly the same. We also make it clear that the baseline and final evaluation survey items were exactly the same.

4. Please discuss: Why almost only women participated? How could you get also men to participate?

 We have discussed this issue as a limitation in the Limitations section that was added.

5. Line 178: 21 out of 25 is 84%. If authors want to use 91% it should be written out that it means 21 out of 23.

We have made it clear that the 91% is based on 21 of 23 Fellows participating.

6. Author name "Carlette Lewis Rhone" is written with bigger font.

This has been corrected.

In addition, the formatting errors, including in the references have been corrected.

Reviewer 3 Report

ijerph-2048726

Title: Community Research Fellows Training Program: Evaluation of a COVID-19 Precipitated Virtual Adaptation

Comments to the Author:

Thank you for your manuscript which clearly describes the evaluation of a program to increase research knowledge and capacity among community partners. 

Major concerns:

·       Please provide more detail related the training of participants which is mentioned on p. 8 line 202.

·       Please provide a discussion on the reach of your program. I noted that all of the online participants had some college experience and over 2/3 had taken a research course… that makes for a very engaged audience that has context for the research process. I would be very interested in a discussion on how to reach those individuals with a high school diploma or less education.

Minor concerns/suggestions:

·       Please consider adding more subheadings particularly to the Methods section to assist your reader.

·       Table 1 is helpful understanding the breadth and depth of your curriculum but adding a column related to activity learning strategies (breakout rooms for small group discussion, role plays, group message boards like Padlet or Jam Board, etc.) for each session would help the reader understand how you adapted to the virtual environment.

·       Please consider the formatting for your tables and possibly using left justification for the left most column.

Author Response

Thank you for your feedback and recommendations. We have addressed each of your suggestions as indicated below. We believe that we have been responsive and that your input has strengthened the paper. 

1. Please provide more detail related the training of participants which is mentioned on p. 8 line 202.

We have provided a paragraph that describes the training that participants received on the LMS systemThis paragraph appears in the Methods section (p. 4; lines 157-175)

2. Please provide a discussion on the reach of your program. I noted that all of the online participants had some college experience and over 2/3 had taken a research course… that makes for a very engaged audience that has context for the research process. I would be very interested in a discussion on how to reach those individuals with a high school diploma or less education.

We have discussed the reach of the program in the limitations section. We note that we have had participants with less education in previous cohorts that were in person. However, the numbers have been very small. We suggest changes in format that may assist in making the program more attractive for individuals with less education.

Minor concerns/suggestions:

Please consider adding more subheadings particularly to the Methods section to assist your reader.

We have added subheadings to the Methods section, as well as a limitations section to the discussion.

3. Table 1 is helpful understanding the breadth and depth of your curriculum but adding a column related to activity learning strategies (breakout rooms for small group discussion, role plays, group message boards like Padlet or Jam Board, etc.) for each session would help the reader understand how you adapted to the virtual environment.

We have provided information on the use of breakout rooms, homework, activities and how they were accomplished.

4. Please consider the formatting for your tables and possibly using left justification for the left most column.

We have left justified the tables as suggested.